# Distributed Representations of Graphs for Drug Pair Scoring

**Paul Scherer, Pietro Liò** and **Mateja Jamnik**
Department of Computer Science and Technology
University of Cambridge
`paul.scherer@cl.cam.ac.uk`

## Abstract

In this paper we study the practicality and usefulness of incorporating distributed representations of graphs into models within the context of drug pair scoring. We argue that the real world growth and update cycles of drug pair scoring datasets subvert the limitations of transductive learning associated with distributed representations. Furthermore, we argue that the vocabulary of discrete substructure patterns induced over drug sets is not dramatically large due to the limited set of atom types and constraints on bonding patterns enforced by chemistry. Under this pretext, we explore the effectiveness of distributed representations of the molecular graphs of drugs in drug pair scoring tasks such as drug synergy, polypharmacy, and drug-drug interaction prediction. To achieve this, we present a methodology for learning and incorporating distributed representations of graphs within a unified framework for drug pair scoring. Subsequently, we augment a number of recent and state-of-the-art models to utilise our embeddings. We empirically show that the incorporation of these embeddings improves downstream performance of almost every model across different drug pair scoring tasks, even those the original model was not designed for. We publicly release all of our drug embeddings for the DrugCombDB, DrugComb, DrugbankDDI, and TwoSides datasets.

## 1 Introduction

Recent advancements in graph representation learning (GRL) — particularly in message passing based graph neural networks — have enabled new ways of modelling natural phenomena and tackling learning tasks on graph structured data. One of the areas which now sees application of graph neural networks is drug pair scoring [1]. Drug pair scoring refers to the prediction tasks that answer questions about the consequences of administering a pair of drugs at the same time such as drug synergy prediction, polypharmacy prediction, and predicting drug-drug interaction types which are of great interest in the treatment of diseases. One of the primary challenges in elucidating and discovering the effects of drug combinations is the dramatically growing combinatorial space of drug pairs. Furthermore, reliance on human trials (in polypharmacy), and proneness to human error [2] makes manual/experimental discovery of useful drug combinations difficult without even considering the prohibitive financial and labour costs that make it only possible on small sets of drugs. Such conditions make *in silico* modelling of drug combinations an attractive solution.

A key component to modelling drug pairs is finding useful representations of the drugs to input into the drug pair scoring models. Traditional supervised machine learning methods for drug pair scoring rely on carefully crafted *descriptors* such as MDL descriptor keysets [3] and fingerprinting techniques such as Morgan fingerprinting [4]. More recently, graph neural network layers and permutation invariant pooling operators have enabled inputting the molecular graphs of drugs directly to learn task oriented representations in an end-to-end manner. Interestingly, graph kernel techniques and specifically distributed representations of graphs were not considered at all for inclusion in drug pair scoring pipelines to the best of our knowledge. We may only speculate to the reasons for this such as publication biases or its limitations in not using node feature vectors and the transductive nature that have made these approaches less appropriate in observations with rich/continuous node features and dynamic graphs [5, 6].

P. Scherer et al., Distributed Representations of Graphs for Drug Pair Scoring. *Proceedings of the First Learning on Graphs Conference (LoG 2022)*, PMLR 198, Virtual Event, December 9–12, 2022.

However, we will argue that the transductive learning of distributed representations is hardly a limitation in the context of drug pair scoring tasks in Section 3.2. This is primarily as we are learning the representations of the drugs whose number in the real world rises in the timescale of many years and immense investment [7, 8]. Furthermore, as the set of atom types and bonding patterns of drugs are strictly constrained by the rules of chemistry, the number of generic substructure patterns that may be induced over the molecular graphs of a drug set are much smaller than the theoretically possible set of combinations. Additionally, as the self supervised learning objective is agnostic to the downstream task the drug embeddings may be transferred trivially making distributed representations an attractive modelling proposition for representation learning of structural patterns for drug pair scoring.

Under this pretext our research questions are: "How can we learn and then incorporate the distributed representations of the drugs into drug pair scoring pipelines?" and "Are distributed representations of graphs useful in drug pair scoring tasks?". To answer these questions we describe a methodology for learning distributed representations of graphs and their inclusion within a unified framework applicable all drug pair scoring tasks in Section 3. Subsequently, we create a simple MLP model based solely on the distributed representations of the drugs and show that this performs considerably better than random suggesting the usefulness of discrete substructure affinities of the drugs in drug pair scoring. Building upon this, we augment a number of recent and state-of-the-art models for drug pair scoring tasks to utilise our drug embeddings. Empirical results show that the incorporation of the distributed representations improves the performance of almost every model across synergy, polypharmacy, and drug interaction prediction tasks in Section 5. To the best of our knowledge this is the first application and study of distributed representations of molecular drug graphs for drug pair scoring tasks. To help further research and inclusion of these distributed representations we publicly release all of the drug representations as learned and utilised in this study.

To summarise **our contributions** are as follows:

- We show that learning distributed representations of graphs as a source of additional features is reasonable within drug pair scoring pipelines.
- We present a generic methodology for learning various distributed representations of the molecular graphs of the drugs and incorporating these into machine learning pipelines for drug pair scoring.
- We augment state-of-the-art models for drug synergy, polypharmacy, and drug interaction prediction and improve their performance through the use of distributed drug representations across tasks; even tasks they were not originally designed for.
- We publicly release all of the drug embeddings for DrugCombDB [2], DrugComb [9, 10], DrugbankDDI [11], and TwoSides [12] datasets as utilised in this study with the accompanying code for generating more.

## 2 Background and Related Work

In drug pair scoring tasks we are concerned with learning a function which predicts scores for pairs of drugs in a biological or chemical context. Naturally, within the domain of deep learning this learned function takes on the form of a neural network. Drug pair scoring have three main applications and questions which models are designed to answer [1]:

- Inferring drug synergy: Do drugs $i$ and $j$ have a synergistic effect on treatment of disease $k$?
- Inferring polypharmacy side effects: Does the simultaneous use of drugs $i$ and $j$ have a propensity for causing side effect $k$?
- Inferring drug-drug interaction types: Do drugs $i$ and $j$ have a $k$ type interaction?

### 2.1 Unified Framework for Drug Pair Scoring

The machine learning tasks born out of the questions above can be generalised and formalised with a unified view of drug pair scoring described in Rozemberczki et al. [1]. We briefly reiterate this framework below to build upon in our proposed work in the next section.

Assume there is a set of $n$ drugs $\mathcal{D} = \{d_1, d_2, ..., d_n\}$ for which we know the chemical structure of molecules and a set of classes $\mathcal{C} = \{c_1, c_2, ..., c_p\}$ that provides information on the contexts under which a drug pair can be administered.

A **drug feature set** is the set of tuples $(\mathbf{x}^d, \mathcal{G}^d, \mathbf{X}_N^d, \mathbf{X}_E^d) \in \mathcal{X}_\mathcal{D}, \forall d \in \mathcal{D}$, where $\mathbf{x}^d$ is the molecular feature vector, $\mathcal{G}^d$ is the molecular graph of the drug, $\mathbf{X}_N^d$ is the node/atom feature matrix and $\mathbf{X}_E^d$ the edge/bond feature matrix. In this setup, drugs can be attributed with 4 types of information: (i) Molecular features which give high-level information about the molecules such as measures of charge. (ii) The molecular graph in which nodes are atoms and edges describe bonding patterns. (iii) Node features in the molecular graph can give us information such as the type of atom or whether it is in a ring. (iv) Edge features which can provide context such as the type of bond that exists between atoms in the molecule.

A **context feature set** is the set of context feature vectors $\mathbf{x}^c \in \mathcal{X}_\mathcal{C}, \forall c \in \mathcal{C}$ associated with the context classes $\mathcal{C}$. This set allows for making context specific presdictions that take into account the similarity of the contexts. For example, in a synergy prediction scenario the context features can describe the gene expressions in a targeted cancer cell.

The **labeled drug-pair and context triple set** is a set of tuples $(d, d', c, y^{d,d',c}) \in \mathcal{Y}$ where $d, d' \in \mathcal{D}$, $c \in \mathcal{C}$ and $y^{d,d',c} \in \{0, 1\}$. This set of observations associates a drug pair within a specific biological or chemical context with a binary target. This target could specify whether a pair of drugs is synergistic in terminating a cancer cell type or have a certain drug-drug interaction type. Naturally, it is also common to have continuous targets $y^{d,d',c} \in \mathbb{R}$. The machine learning practitioner is tasked with constructing predictive models $f(\cdot)$ such that $\hat{y}^{d,d',c} = f(d, d', c)$ for these drug-pair context observations.

## 2.2 Representations for Drugs

A major source of research interest is the study and development of drug feature vectors and representations as they form inputs into various drug learning tasks. In our case these form integral parts of the molecular feature vector $\mathbf{x}^d$ in the drug feature set (see Section 2.1) often arising from the molecular graph of the drugs.

Two dimensional representations and diagrams of the structure of molecules are often used as a convenient representation for their 3-dimensional structures and electrostatic properties that give rise to their biological activities. Whilst this abstraction is useful for communication in person, technical limitations drove the development of linear string based representations including SMILES [13] and InChI [14] which are present across many popular chemical information systems today. Language models have been applied onto such molecular strings to learn embeddings such as in Bombarelli et al. [15] which utilises the SMILES strings within a VAE framework to sample low dimensional continuous vector representations of the drugs. The success of this inspired similar work such as DeepSMILES [16] and SELFIES [17].

Two dimensional graph structures have been used before to generate discrete bag-of-words type feature vectors of molecules based on the presence of a specified vocabulary of descriptive substructures as in Morgan's work in 1965 [18]. Subsequent years saw efforts in finding different descriptive properties within the molecule structures or optimising existing sets of descriptive substructures such as in Durant et al. [3] which optimised the set of substructure based 2D descriptors from MDL keysets for drug discovery pipelines. The use of molecular fingerprints such as Morgan/Circular fingerprints [4] continues this branch of constructing descriptors and kernels for molecules. Concurrent efforts recently focus on end-to-end neural models involving graph neural network operators [1, 19]. Here graph neural networks operate over the molecular graph of the drug such that atoms are treated as nodes and bonds are the edges. Node level representations are updated through a series of message passing layers as in Equation 1 as described in Gilmer et al. [20] and Battaglia et al. [21].

$$\mathbf{h}_i^l = \phi\big(\mathbf{h}_i^{l-1}, \bigoplus_{j \in \mathcal{N}_i} \psi(\mathbf{h}_i^{l-1}, \mathbf{h}_j^{l-1})\big) \tag{1}$$

Here $\mathbf{h}_i^l$ is the $l$th layer representation of the features associated with node $i$ (in our context these would be atom features arising from message passing using $\mathbf{X}_N^d$ and $\mathbf{X}_E^d$). $\mathbf{h}_i^l$ is the output of the local permutation invariant function composed of the node $i$'s previous feature representation $\mathbf{h}_i^{l-1}$ and its neighbours $j \in \mathcal{N}_i$ with $\psi(\mathbf{h}_i^{l-1}, \mathbf{h}_j^{l-1})$ being the message computed via function $\psi$ and $\bigoplus$ is some permutation invariant aggregation for the messages such as a sum, product, or average. $\phi$ and $\psi$ are typically neural networks. Subsequently, the node level representations are aggregated via

permutation invariant pooling operations to form graph-level drug representations. For example, the EPGCN-DS model [22] utilises GCN layers [23] to produce higher level node representations of the atoms in the molecular graphs. The drug representations are then computed via a mean aggregation of the node representations. Such operators have become prevalent in recent proposals of drug pair scoring models with primary distinction being the form of $\psi$ in the message passing layers [1, 22, 24, 25].

Our proposed system lies somewhere in between and in parallel to these efforts. We learn low dimensional continuous distributed representations (described in Section 3.1) of the drugs within the drug pair scoring dataset. These form additional drug features that can be utilised in augmented versions of existing drug pair scoring models. To the best of our knowledge this is the first application of distributed representations of drugs within drug pair scoring.

## 2.3    Neural Models for Drug Pair Scoring

All recent neural models for drug pair scoring can be described with an encoder-decoder framework typically involving 3 parametric functions: (i) a drug encoder, (ii) an encoder for contextual features, and (iii) a decoder which infers the target value. We describe each component below, followed by how some state-of-the-art models can be instantiated out of this framework. A more thorough treatment of this can be found in Rozemberczki et al. [1].

The drug encoder is the parametric function $f_{\theta_D}(\cdot)$ in Equation 2 that takes the drug feature set as input and produces a vector representation of the drug $d$ called $\mathbf{h}^d$. $f_{\theta_D}(\cdot)$ maps the molecular features of the drug into a low dimensional vector space, this can incorporate various neural operators such as feed forward multi-layer perceptron layers as in DeepSynergy [26] and MatchMaker [27] or graph neural network layers as in DeepDDS [24] and DeepDrug [25]. Differences in the architecture of the encoder such as the flavour of message passing network is typically the main differentiator between current existing methods.

$$\mathbf{h}^d = f_{\theta_D}(\mathbf{x}^d, \mathcal{G}^d, \mathbf{X}_N^d, \mathbf{X}_E^d), \forall d \in \mathcal{D} \tag{2}$$

The context encoder $f_{\theta_C}(\cdot)$ in Equation 3 is a neural network that outputs a low dimensional representation of the contextual feature set $\mathbf{x}^c$. This component does not feature in all of the models we will discuss but plays a prominent part in DeepSynergy [26], MatchMaker [27], and DeepDDS [24].

$$\mathbf{h}^c = f_{\theta_C}(\mathbf{x}^c), \forall c \in \mathcal{C} \tag{3}$$

Finally the decoder or head of the model $f_{\theta_H}(\cdot)$ in Equation 4 combines the outputs of the drug and context encoders $(\mathbf{h}^d, \mathbf{h}^{d'}, \mathbf{h}^c)$ and outputs the predicted probability for a positive label for the drug-pair context triple $\hat{y}^{d,d',c}$.

$$\hat{y}^{d,d',c} = f_{\theta_H}(\mathbf{h}^d, \mathbf{h}^{d'}, \mathbf{h}^c), \forall d, d' \in \mathcal{D}, \forall c \in \mathcal{C} \tag{4}$$

Training the models in the framework described involves minimising the binary cross entropy for the binary targets or mean absolute error for regression targets with respect to the $\theta_D$, $\theta_C$, and $\theta_H$ parameters using gradient descent algorithms.

$$\mathcal{L} = \sum_{(d,d',c,y^{d,d',c}) \in \mathcal{Y}} l(\hat{y}^{d,d',c}, y^{d,d',c}) \tag{5}$$

## 3    Study and Methods

### 3.1    Distributed Representations of Graphs

We adopt the framework of Scherer and Liò [28] for describing distributed representations of graphs based on the R-Convolutional framework for graph kernels [29]. Given a set of $n$ molecular graphs for the drugs in the dataset $\mathbb{G} = \{\mathcal{G}^{d_1}, \mathcal{G}^{d_2}, ..., \mathcal{G}^{d_n}\}$ one can induce discrete substructure patterns such as shortest paths, rooted subgraphs, graphlets, etc. using side effects of algorithms such as

Floyd-Warshall [30–32] or the Weisfeiler-Lehmann graph isomorphism test [33]. This can be used to produce pattern frequency vectors $X = \{x^{d_1}, x^{d_2}, ..., x^{d_n}\}$ describing the occurence frequency of substructure patterns for every graph over a shared vocabulary $\mathbb{V}$. $\mathbb{V}$ is the set of unique substructure patterns induced over all graphs $\mathcal{G}^d \in \mathbb{G}$.

Classically one may directly use these pattern frequency vectors within standard machine learning algorithms or construct kernels to perform some task. This has been the approach taken by many state of the art graph kernels in classification tasks [29, 34]. Unfortunately, as the number, complexity, and size of graphs in $\mathbb{G}$ increases so does the number of induced substructure patterns — often dramatically [28, 29, 34]. This, in turn, causes the pattern frequency vectors of $X$ to be extremely sparse and high dimensional both of which are detrimental to the performance of estimators. Furthermore, the high specificity of the patterns and the sparsity cause a phenomenon known as diagonal dominance across kernel matrices wherein each graph becomes more similar to itself and dissimilar from others, degrading machine learning performance.

To address this issue it is possible to learn dense and low dimensional distributed representations of graphs that are inductively biased to be similar when they contain similar substructure patterns and dissimilar if they do not in a self supervised manner. To achieve this we need to construct a corpus dataset $\mathcal{R}$ that details the target-context relationship between a graph and its induced substructure patterns. In the simplest form for graph level representation learning we can specify $\mathcal{R}$ as the set of tuples $(\mathcal{G}^d, p) \in \mathcal{R}$ where $p$ is a substructure pattern that is part of the shared vocabulary $p \in \mathbb{V}$ and can be induced from $G^d$ which we denote $p \in \mathcal{G}^d$.

The corpus can then be used to learn embeddings via a method that incorporates Harris' distributive hypothesis [35] to learn the distributed representations. Methods such as Skipgram, CBOW, PV-DM, PV-DBOW, and GLoVE are some examples of neural embedding methods that utilise this inductive bias [36–38]. In our study we implement Skipgram with negative sampling which optimises the following objective function.

$$\mathcal{L} = \sum_{\mathcal{G}^d \in \mathbb{G}} \sum_{p \in \mathbb{V}} |\{(\mathcal{G}^d, p) \in \mathcal{R}\}|(\log \sigma(\Phi^d \cdot \mathcal{S}_p)) + \mathbb{E}_{p^- \in \mathbb{V}}[\log \sigma(-\Phi^d \cdot \mathcal{S}_{p^-})] \tag{6}$$

Here $\mathbf{\Phi} \in \mathbb{R}^{|\mathbb{G}| \times z}$ is the $z$-dimensional matrix of graph embeddings we desire of the set of drug graphs $\mathbb{G}$, and $\Phi^d$ is the embedding for $\mathcal{G}^d \in \mathbb{G}$. In similar vein, $\mathcal{S} \in \mathbb{R}^{|V| \times z}$ are the $z$-dimensional embeddings of the substructure patterns such that $\mathcal{S}_p$ represents the vector embedding corresponding to the substructure pattern $p \in \mathbb{V}$. Whilst these embeddings are tuned as well during the optimisation of Equation 6, ultimately, these substructure embeddings are not used in our case as we are interested in the drug embeddings. The cardinality of the set $|\{(\mathcal{G}^d, p) \in \mathcal{R}\}|$ indicates the number of time a positive substructure pattern is induced in the graph to tighten the association of the pattern to the graph. $p^- \in \mathbb{V}$ denotes a negative context pattern that is drawn from the empirical unigram distribution $P_R(p) = \frac{|\{p|\forall \mathcal{G}^d \in \mathbb{G}, (\mathcal{G}^d, p) \in \mathcal{R}\}|}{|\mathcal{R}|}$ and the expectation is approximated using 10 Monte Carlo samples as originally devised in Mikolov et al. [36].

The optimisation of the above objective creates the desired distributed representations in $\mathbf{\Phi}$, in this the case graph-level drug embeddings. These may be used as additional drug features in the drug feature set as we show in section 3.3. The distributed representations benefit from having lower dimensionality than the pattern frequency vectors, in other words $|V| >> z$, being non-sparse, and being inductively biased via the distributive hypothesis. A more thorough treatment of the distributive hypothesis and in-depth interpretation of the embedding methods in this family can be found in [35, 36, 39].

Various instances of models for learning distributed representations of graphs following our description have been made such as Graph2Vec [40], DGK-WL/SP/GK [29], and AWE [41]. These differentiate primarily on the type of substructure pattern is induced over $\mathbb{G}$. These have shown strong performance in graph classification tasks, still often performing on par with modern graph neural networks despite using significantly less features and parameters. However, limitations such as the dependency on a set vocabulary and inability to inductively infer representations for new subgraph patterns and new graphs (at least in its standard definitions), coupled with difficulty in scaling to large graphs with many millions of node have led to less attention on these methods. We speculate this has

**Table 1:** Dataset details containing information on the application domain, and summary statistics on the number of drugs, context types, and drug pair context triples. Additional columns highlight the number of unique substructure patterns found across the molecular graphs of the drugs in the dataset based on the substructure patterns induced. $|\mathcal{D}|$ represents the number of unique drugs. $|\mathcal{C}|$ represents the set of unique contexts. $|\mathcal{Y}|$ represents the number of labeled drug-drug context triples. The remaining columns indicate the number of unique substructure patterns found in the drugs with respect to the corresponding substructure patterns extracted: WL ($k = 2$) is the number of discrete rooted subtrees up to depth 2, WL ($k = 3$) for rooted subgraphs up to depth 3, and the shortest paths.

| Dataset | Task | $|\mathcal{D}|$ | $|\mathcal{C}|$ | $|\mathcal{Y}|$ | WL ($k = 2$) | WL ($k = 3$) | Shortest paths |
|---|---|---|---|---|---|---|---|
| DrugCombDB [2] | Synergy | 2956 | 112 | 191,391 | 70 | 1591 | 1310 |
| DrugComb [9, 10] | Synergy | 4146 | 288 | 659,333 | 70 | 1651 | 1432 |
| DrugbankDDI [11] | Interaction | 1706 | 86 | 383,496 | 74 | 1287 | 2710 |
| TwoSides [12] | Polypharmacy | 644 | 10 | 499,582 | 64 | 934 | 8070 |

led to developments of deep drug pair score models completely ignoring distributed representations of graphs as part of the pipeline.

## 3.2 Arguing for the Use of Distributed Representations of Drugs in Drug Pair Scoring Pipelines

Here we show that the use of distributed representations of graphs to construct additional drug features is sensible in drug pair scoring tasks. As discussed in Section 2.1 a drug score pairing model is tasked with learning the function $f(d, d', c) = y^{d,d',c}$ from the labelled drug-pair context triples in $\mathcal{Y}$. Looking at the statistics of drug pair scoring datasets in Table 1, we can see that the number of drugs and contexts is far lower than the number of triple observations. The huge and complex combinatorial space of drug-pair contexts (without even considering dosage effects) as well as the time/cost associated with experimenting more triples is a motivating factor for machine learning models. In practice, when such databases are updated it is through the addition of more labelled drug-pair context observations for better coverage [42]. The number of drugs considered rarely increases, as drugs can take many years of development, clinical trials, massive investment and regulatory processes before they enter studies for application domains of drug pair scoring [7, 8].

Therefore we can argue that learning distributed representations of the molecular graphs of the drugs in drug pair scoring tasks is sensible. Importantly, the number of discrete substructure patterns grows with the number of unique drugs, not the number of drug-pair-context observations within the dataset. Hence, as long as the number of drugs stays the same, trained drug embeddings can be carried over to any model being trained over the drug-pair context triples with minimal augmentation as we show in Section 3.3. To add further motivation, the number of discrete substructure patterns in the considered set of drugs is driven by the unique atom types and substructure patterns arising out of the bonded atoms. This set of unique atom types is theoretically limited to the periodic table and is obviously a limited subset of this in drugs. Furthermore, the size of the molecular graphs tend to be considerably smaller than social network scale graphs and less random due to chemical bonding rules hence the resulting substructure patterns are fewer and more informative making suitable descriptors in these settings [29, 34, 43].

## 3.3 Incorporating Distributed Representations of Graphs into Existing Drug Pair Scoring Pipelines

Through retrieval of the SMILES strings, we generated the molecular graphs for each of the drugs $\mathbb{G} = \{\mathcal{G}^d | d \in \mathcal{D}\}$ using TorchDrug [44] and RDKit [45]. Given this set of graphs we considered two discrete substructure patterns to induce over the graphs. For the first substructure pattern we considered rooted subgraphs at different depth $k = 3$. These may be induced as a side effect of the Weisfeiler-Lehman graph isomorphism test [33, 43]. The second substructure pattern we considered were all the shortest paths of the molecular graph which may be induced using the Floyd-Warshall algorithm [30–32]. Both choices were made based on their completeness and deterministic nature of their inducing algorithms for which there are also fast implementations [28, 46].

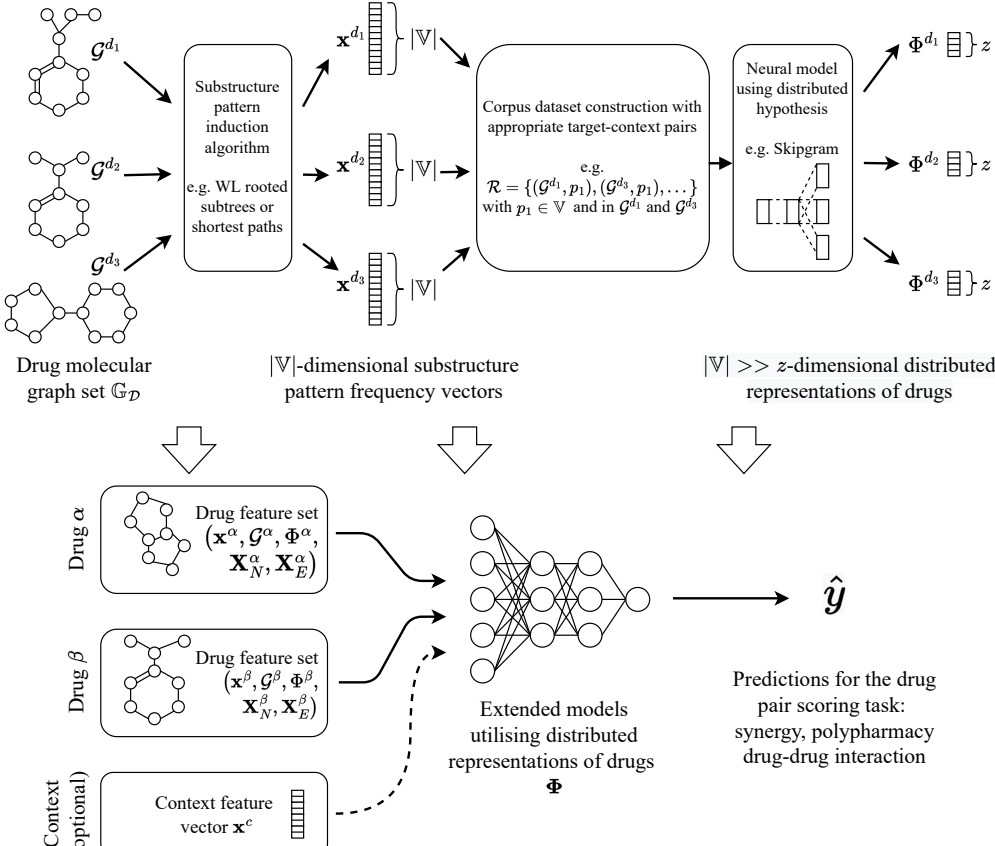

**Figure 1:** A summary of the proposed pipeline for learning and utilising distributed representations of drugs for drug pair scoring. The pipeline consists of two main stages: the learning of the distributed representations and the augmentation of existing models to utilise the new drug embeddings $\mathbf{\Phi}$ which become part of the drug feature set described in Section 2.1. As the learning of the distributed representations is separate from the drug pair scoring task we may transfer the embeddings into the drug feature set of any existing drug pair scoring model without retraining.

In either case, the set of unique substructure patterns found across all molecular graphs in $\mathcal{D}$ gives us the molecular substructure vocabulary $\mathbb{V}$. We construct a target-context corpus of the drugs $\mathcal{R}_{\mathcal{D}} = \{(\mathcal{G}^d, p) | \mathcal{G}^d \in \mathbb{G}, p \in \mathcal{G}^d, p \in \mathbb{V}\}$. We use a skipgram model with negative sampling to learn the desired drug embeddings, optimising the objective function in equation 6.

After training and obtaining the distributed representations of drugs $\mathbf{\Phi}$ we add the embeddings to the drug feature set $(\mathbf{x}^d, \mathbf{\Phi}^d, \mathcal{G}^d, \mathbf{X}_N^d, \mathbf{X}_E^d) \in \mathcal{X}_{\mathcal{D}}, \forall d \in \mathcal{D}$. The remaining task is to develop downstream models which utilise the distributed representations. As the self supervised learning of the distributed representations is separate from the learning for the drug pair scoring task, we may transfer the embeddings into any of the existing drug pair scoring models. A diagram of this workflow can be seen in Figure 1.

In order to validate the usefulness of the distributed representations we chose to extend existing drug pair scoring models from different application domains. As a sanity check to see whether the distributed representations carry any useful signal we also implemented a simple MLP with three hidden layers based on DeepSynergy called DROnly which only utilises the embeddings learned. We took seminal models representing the state of the art and recent models containing graph neural networks that operate over the molecular graphs of the drugs. Each augmented model we propose takes the original name of the model and is suffixed with "DR" and the substructure pattern induced over the graphs (WL or SP for rooted subgraphs and shortest paths respectively). In most cases we simply concatenate the distributed representation of the first and second drug (drugs $\alpha$ and $\beta$ in Figure 1) to the corresponding molecular feature vectors being used in the model. In the case

of EPGCN-DS-DR and DeepDrugDR the left and right drug embeddings are concatenated to the outputs of the graph neural network drug encoders and fed into the decoder.

## 4   Experimental Setup

We empirically validate the usefulness of the distributed drug representations in downstream drug pair scoring tasks. We consider 4 datasets from the domains of drug synergy prediction, polypharmacy prediction, and drug interaction to evaluate our augmented models, which we have previously outlined in Table 1. Five seeded random 0.5/0.5 train and test set splits were made and the average AUROC performance was evaluated over the hold-out test set with standard deviation in Table 2.

For the distributed representations of the graphs we set the desired dimensionality at $z = 64$ and the Skipgram model was trained for 1000 epochs. These hyperparameter values were chosen arbitrarily to simplify the following comparative analysis, however we explore their effects on downstream performance in an ablation study in Appendix A.

To obtain the non DR drug-level features as used in DeepSynergy and MatchMaker we retrieved the canonical SMILES strings [13] for each of the drugs in the labeled drug-pair context triples. 256 dimensional Morgan fingerprints [4] were computed for each drug with a radius of 2. Molecular graphs for entry into models with GNNs were generated using TorchDrug (and the underlying RDKit utilities) from the SMILES strings for each drug.

We utilised the default hyperparameters for each of the drug pair scoring models as in [47] which are summarised in Table 3 of Appendix B. Augmentation of the models affects the input shapes of the drug encoders or the final decoder by the chosen dimensionality of the distributed representations, but does not affect any other original model hyperparameters.

Optimisation hyperparameters for training of the models were all kept the same. All drug pair scoring models were trained using an Adam optimiser [48] for 250 epochs with a batch size of 8192 observations, an initial learning rate of $10^{-2}$, $\beta_1$ was set to 0.9 with $\beta_2$ set to 0.99, $\epsilon = 10^{-7}$ and finally a weight decay of $10^{-5}$ was added. A dropout rate of 0.5 was applied for regularisation.

Naturally in addition to these details we make all of our code containing all implementations and scripts for evaluation available on https://github.com/paulmorio/DrugPairScoringDR for reproducibility and further development.

## 5   Results and Discussion

Looking at our main results Table 2 we can make 3 main observations. First looking at the original methods we can see that methods using precomputed drug features and contextual features instead of graph neural networks such as DeepSynergy and Matchmaker perform better across drug pair scoring tasks. Combined with the fact that they train and evaluate much faster than methods using graph neural networks, it is generally advisable to use these models in the first instance validating the results in [47]. DeepDDS is the best performing model utilising a graph neural network. It is worth noting that it utilises contextual features like DeepSynergy and MatchMaker and unlike EPGCN-DS and DeepDrug. Secondly, looking at the DROnly model that serves as the sanity check for our embeddings, we can see that it is significantly better than a random model. This indicates the usefulness of the structural affinities and distributive inductive biases within the drug representations for the drug pair scoring tasks. Thirdly, we can see that the incorporation of the distributed representation into the models generally increases the performance of models. Particularly, we observe that the best performances for 3 out of 4 tasks are achieved by models incorporating our embeddings with the final one being a tie (within rounding error of 3 decimal points) between DeepDDS and its DR incorporating equivalent DeepDDS-DR (WL k=3) on TwoSides.

The horizontal analysis of the drug pair scoring models highlights that the significantly more expensive graph neural network based models generally perform worse than simpler models employing precomputed drug and context features on MLPs. This in spite of the graph neural network modules also having access to additional atom features on the molecular graphs as computed in TorchDrug. These include features such as the one-hot embedding of the atomic chiral tag, whether it participates in a ring, and whether it is aromatic, and the number of radical electrons on the atom. Hence, despite the wealth of additional information inside the provided molecular graph, we surmise the primary

**Table 2:** Table of results with information about the original drug pair scoring models such as year of publication and their original application domains. We report the average AUROC on the hold out test set with standard deviations from 5 seeded random splits. Bolded numbers indicate best performing model for each dataset.

| Model | Year | Orig. application | DrugCombDB | DrugComb | DrugbankDDI | TwoSides |
|---|---|---|---|---|---|---|
| DeepSynergy [26] | 2018 | Synergy | 0.796 +- 0.010 | 0.739 +- 0.005 | 0.987 +- 0.001 | 0.933 +- 0.001 |
| EPGCN-DS [22] | 2020 | Interaction | 0.703 +- 0.006 | 0.623 +- 0.002 | 0.724 +- 0.002 | 0.809 +- 0.006 |
| DeepDrug [25] | 2020 | Interaction | 0.743 +- 0.001 | 0.648 +- 0.001 | 0.862 +- 0.002 | 0.926 +- 0.001 |
| DeepDDS [24] | 2021 | Synergy | 0.791 +- 0.005 | 0.697 +- 0.002 | 0.988 +- 0.001 | **0.944 +- 0.001** |
| MatchMaker [27] | 2021 | Synergy | 0.788 +- 0.002 | 0.720 +- 0.003 | 0.991 +- 0.001 | 0.928 +- 0.001 |
| DROnly (WL k=3) | Proposed | Not applicable | 0.763 +- 0.002 | 0.651 +- 0.002 | 0.809 +- 0.005 | 0.917 +- 0.002 |
| DROnly (SP) | Proposed | Not applicable | 0.711 +- 0.004 | 0.621 +- 0.002 | 0.710 +- 0.005 | 0.823 +- 0.005 |
| DeepSynergy-DR (WL k=3) | Proposed | Not applicable | **0.814 +- 0.004** | 0.738 +- 0.001 | 0.988 +- 0.000 | 0.934 +- 0.002 |
| DeepSynergy-DR (SP) | Proposed | Not applicable | 0.813 +- 0.003 | **0.740 +- 0.004** | 0.988 +- 0.001 | 0.935 +- 0.000 |
| EPGCN-DS-DR (WL k=3) | Proposed | Not applicable | 0.711 +- 0.002 | 0.627 +- 0.001 | 0.741 +- 0.004 | 0.822 +- 0.006 |
| EPGCN-DS-DR (SP) | Proposed | Not applicable | 0.704 +- 0.001 | 0.622 +- 0.001 | 0.730 +- 0.003 | 0.808 +- 0.002 |
| DeepDrug-DR (WL k=3) | Proposed | Not applicable | 0.743 +- 0.001 | 0.648 +- 0.001 | 0.863 +- 0.001 | 0.926 +- 0.001 |
| DeepDrug-DR (SP) | Proposed | Not applicable | 0.743 +- 0.000 | 0.648 +- 0.001 | 0.863 +- 0.001 | 0.926 +- 0.000 |
| DeepDDS-DR (WL k=3) | Proposed | Not applicable | 0.799 +- 0.004 | 0.700 +- 0.002 | 0.989 +- 0.000 | **0.944 +- 0.001** |
| DeepDDS-DR (SP) | Proposed | Not applicable | 0.790 +- 0.003 | 0.696 +- 0.001 | 0.988 +- 0.001 | 0.943 +- 0.001 |
| MatchMaker-DR (WL k=3) | Proposed | Not applicable | 0.783 +- 0.004 | 0.714 +- 0.003 | **0.992 +- 0.000** | 0.930 +- 0.001 |
| MatchMaker-DR (SP) | Proposed | Not applicable | 0.784 +- 0.002 | 0.714 +- 0.004 | 0.991 +- 0.001 | 0.928 +- 0.002 |

bottleneck for the drug level representations arises from the comparatively simple permutation invariant operators used to pool the node representations such as the global mean operator used in EPGCN-DS. There is an inevitable and large amount of information loss in the attempt to summarise variable amounts of higher level smooth node representations coming out of GNNs into a single vector of the same size, without any trainable parameters. We may partially attribute the additional performance boosts brought in by the distributed representations to the more refined algorithm to constructing the graph level representations, despite the input molecular graph only detailing the atom types and no additional node features. We can also attribute the performance boosts to the usefulness of substructure affinities to the drug pair scoring tasks as indicated in the DROnly performances across the tasks.

Appendix C presents two additional experiments which were performed to study the effectiveness of distributed representations in more challenging drug pair scoring scenarios. The first involves constructing a more challenging train-test splits of the drugs and ensuring that a test set of triple observations contains drugs that the model has never seen in training. The second experiment involves studying the effect of distributional shifts in the substructure patterns caused by learning distributed representations over a different superset of drugs to the set found in the dataset and the effect of this on downstream performance. In the latter experiment, we utilised the distributed representations of all unique drugs across the four datasets. In both of these experiments we make the same observations as above and further find positive outcomes of incorporating distributed representations of graphs in drug pair scoring models.

The learning of the distributed representations comes with two hyperparameters which may affect downstream performance when incorporated into the drug pair scoring models. These use specified hyperparameters are: (i) the dimensionality of the drug embeddings and (ii) the number of epochs for which the skipgram model is trained. We give full details on an ablation study on how varying these hyperparameters affects downstream performance with the experimental setup in Appendix A. To summarise the main points: the downstream performance caused by varying the desired dimensionality initially rises and then falls as expected due to the information bottleneck in very small dimensions and curse of dimensionality in higher dimensions. For varying the training epochs we find a slight but statistically significant positive correlation with performance as the number of epochs increases in two out of four datasets. However, in both cases there is little variation ($\pm 0.02$ ROCAUC in both ablation studies over the ranges studied) in the final performance of the downstream models given the hyperparameter choices except on the extreme ends of the studied ranges. This indicates the stable nature of the output embeddings and their usefulness in downstream tasks. As such we can generally recommend low dimensional embeddings on par with any other drug features being utilised and a high number of training epochs to obtain good performance.

# 6 Conclusion

We have answered our two research questions posed in the introduction. We presented a methodology for learning and incorporating distributed representations of graphs into machine learning pipelines for drug pair scoring, answering the first question on how we may integrate distributed representations. We assessed the usefulness of the distributed representations of drugs with two parts. In the first part we show that a model only using the learned drug embeddings shows significantly better performance than random, suggesting the usefulness of the substructure pattern affinities between drugs in drug pair scoring. Subsequently for the second part, we augmented recent and state-of-the-art models from synergy, polypharmacy, and drug interaction type prediction to utilise our distibuted representations. Horizontal evaluation of these models shows that the incorporation of the distributed representations improves performance across different tasks and datasets.

# Acknowledgements

P.S. acknowledges funding by the W.D. Armstrong Fund from the School of Technology at the University of Cambridge. The authors would like to thank Rozemberczki et al. [47] making their software package publicly available. The authors also want to thank the reviewers whose comments led to several refinements and insightful additional experiments for our paper during the review process.

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

## A  Ablation study over the two hyperparameters in learning distributed representations

The introduction of the distributed representations comes with two hyperparameters which may affect their downstream performance when incorporated into the drug pair scoring models. These user specified hyperparameters are: (i) the dimensionality of drug embeddings and (ii) the number of epochs for which the skipgram model is trained. We study the effect of the embedding dimensionality on downstream performance by setting the number of training epochs to 1000 and varying the dimensionality from 8 to 1024 following powers of 2. For our downstream drug pair scoring model we use DeepSynergyDR whilst keeping the same hyperparameter settings as in our comparative analysis described in Section 4. Similarly for studying the effect of training epochs we set the dimensionality of the embeddings at 64 and observe the downstream performance of the drug pair scoring model (with its own training epochs set at 250 as before) across a range of values (from 200 to 2000, in steps of 200). In both cases, we perform 5 repeated runs to obtain empirical confidence intervals in the plots shown in Figures 2 and 3.

### A.1  Dimensionality of distributed representations

The plots in Figure 2 summarise the effects of changing the dimensionality of the drug embeddings on downstream drug pair scoring performance with the DeepSynergyDR model. Across the datasets as well as the substructure patterns we observe there is little change in the downstream performance as the dimensionality increases from 8 to 1024. Furthermore, the resulting downstream performance seems robust against these changes with small confidence regions as in the plots. Both observations suggest that the skipgram model is effective in producing consistent drug gram matrices and capturing salient distributive context information within the embeddings. A Pearson correlation coefficient of 0.331 (p-value: 0.0097) and 0.584 (p-value: $9.873 \times 10^{-7}$) across substructure patterns on DrugCombDB

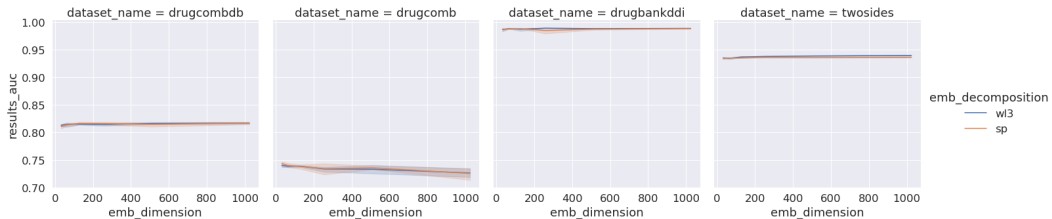

**Figure 2:** Figure of the test ROCAUC performance of DeepSynergyDR (SP and WL3) across the drug pair scoring datasets. Performance is recorded with respect to the embedding dimension chosen in the learning of the distributed representations of graphs.

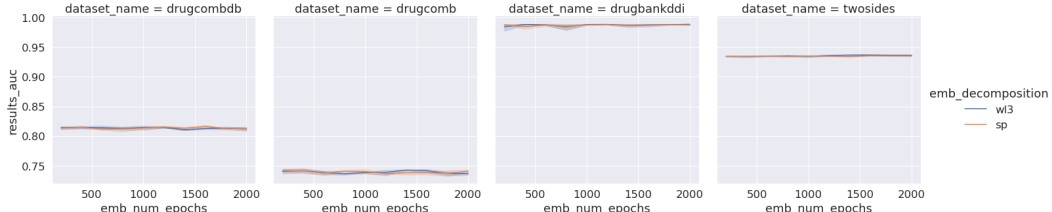

**Figure 3:** Figure of the test ROCAUC performance of DeepSynergyDR (SP and WL3) across the drug pair scoring datasets. Performance is recorded with respect to the number of training epochs chosen in the learning of the distributed representations of graphs.

and TwoSides respectively indicates a statistically significant upward correlation in performance for increased dimensionality. Conversely we find a downwards Pearson correlation coefficient of -0.573 (p-value: $1.692 \times 10^{-6}$) in DrugComb. There is no statistically significant trend (p-value $\leq 0.05$) in DrugbankDDI. Despite the observed upwards trends in performance DrugCombDB and TwoSides we do not recommend having a high embedding dimensionality as we expect an inevitable decrease in performance due to the curse of dimensionality. Hence, we suggest a more moderate choice on par with the dimensionality of other features in the drug feature set as the performance generally is stable across the range of dimensionalities. The next ablation study studies how this varies under the number of training epochs.

## A.2   Number of training epochs for distributed representations

The plots in Figure 3 summarises the effects of changing the number of epochs used in training the skipgram model for a set embedding dimensionality of 64. The plots report the downstream test ROCAUC performance achieved on the DeepSynergy model. Like before, we see that across datasets and induced substructure pattern the downstream performance is not affected strongly except when the number of training epochs is exceptionally low for obvious optimisation reasons. The small confidence bands indicate small variability between different runs. A Pearson correlation coefficient of 0.197 (p-value: 0.049) for DrugbankDDI and 0.473 (p-value: $6.597 \times 10^{-7}$) for TwoSides across substructure patterns indicates a light but statistically significant upwards trend in performance as the number of training epochs increases. DrugcombDB and DrugComb do not show any statistically significant correlations with regard to training epochs, but are generally stable irregardless. Hence we may suggest generally that more rigorous training regimes for learning the distributed representations are favourable in drug pair scoring tasks.

## B   Hyperparameters for the drug pair scoring models

Table 3 summarises the architectural hyperparameters of the drug pair scoring models utilised in this study. Note that these hyperparameters are the same for the DR augmented versions of these models as it only affects the input sizes to the drug encoders (or decoders in the case of EPGCN-DS-DR and DrugDrugDR).

**Table 3:** A breakdown of the hyperparameters in each of the drug pair scoring models. Note that these are the same for each of the augmented versions with distributed representations that we propose.

| Model | Hyperparameter | Values |
|---|---|---|
| DeepSynergy | Drug encoder channels | 128 |
| | Context encoder channels | 128 |
| | Hidden layer channels | (32, 32, 32) |
| EPGCN-DS | Drug encoder channels | 128 |
| | Hidden layer channels | (32, 32) |
| DeepDrug | Drug encoder channels | (32, 32, 32, 32) |
| | Hidden layer channels | 64 |
| DeepDDS | Context encoder channels | (512, 256, 128) |
| | Hidden layer channels | (512, 128) |
| MatchMaker | Drug encoder channels | (32, 32) |
| | Hidden layer channels | (64, 32) |

**Table 4:** Table of dataset details containing information summary statistics on the number of drugs and drug pair context triples based on the train-test splitting procedure detailed in Appendix C.1. $|\mathcal{D}|$ represents the number of unique drugs. $|\mathcal{Y}|$ represents the number of labeled drug-drug context triples. $|A|$ represents the number of unique drugs present across the training set of drug-drug context triples. $|B|$ represents the number of unique drugs present across the test set of drug-drug context triples and are not seen at all during the training process. $|\mathcal{Y}_{train}|$ and $|\mathcal{Y}_{test}|$ represent the number of train and test drug-drug context triples created out of the protocol respectively.

| Dataset | $|\mathcal{D}|$ | $|A|$ | $|B|$ | $|\mathcal{Y}|$ | $|\mathcal{Y}_{train}|$ | $|\mathcal{Y}_{test}|$ |
|---|---|---|---|---|---|---|
| DrugCombDB | 2956 | 2586 | 370 | 191,391 | 113,308 | 78,083 |
| DrugComb | 4146 | 3959 | 187 | 659,333 | 579,891 | 79,442 |
| DrugbankDDI | 1706 | 1298 | 408 | 383,496 | 237,515 | 146,101 |
| TwoSides | 644 | 604 | 40 | 499,582 | 440,718 | 58,864 |

## C  Additional experiments

In addition to our comparative analysis presented in the main part of the paper, two additional experiments were performed to study the effectiveness of distributed representations in more challenging drug pair scoring scenarios. The first involves constructing a more challenging train-test splits of the drugs and ensuring that a test set of triple observations contains drugs that the model has never seen in training. The second experiment involves studying the effect of distributional shifts in the substructure patterns caused by learning distributed representations over a different superset of drugs to the set found in the dataset and the effect of this on downstream performance. *Due to time and hardware constraints this is performed on the MatchMaker and DeepDDS methods (and our DR augmented variants of these) which represent the most recent and state-of-the-art MLP and GNN methods respectively.*

### C.1  Experiments predicting on unseen drugs

To construct a train-test split which ensures that a test set of drug-pair context triple observations contains drugs that the model has never seen in training we performed the following steps:

1. We precomputed a pairwise distance matrix for all of the drugs $d_1, d_2, ..., d_n \in \mathcal{D}$ using the Tanimoto similarity $T(d_i, d_j)$. We used $1 - T(d_i, d_j)$ to get the equivalent distance measure.

2. Split the drugs into two sets $A$ and $B$ using agglomerative clustering with a complete linkage criterion on our Tanimoto based distance matrix to split the drugs $\mathcal{D}$. This ensures that drugs belonging to $A$ are more similar to each other and dissimilar to those in $B$ (and vice versa).

3. Subsequently, for every pair of drugs $(d_i, d_j)$ that make up our observations in the triples we do the following.

**Table 5:** Table of results reporting the average AUROC on the hold out test set that includes drugs that are never seen across any training pair of drugs. We report the average AUROC on the hold out test set with standard deviations from 5 repeated runs. Bolded numbers indicate best performing model for each dataset.

| Model | Year | DrugCombDB | DrugComb | DrugbankDDI | TwoSides |
|---|---|---|---|---|---|
| DeepDDS | 2021 | 0.617 +- 0.010 | 0.573 +- 0.008 | 0.919 +- 0.003 | 0.698 +- 0.035 |
| MatchMaker | 2021 | 0.666 +- 0.015 | 0.580 +- 0.003 | 0.938 +- 0.004 | 0.729 +- 0.018 |
| DROnly (WL k=3) | Proposed | 0.534 +- 0.011 | 0.517 +- 0.005 | 0.636 +- 0.011 | 0.626 +- 0.020 |
| DROnly (SP) | Proposed | 0.542 +- 0.018 | 0.515 +- 0.010 | 0.612 +- 0.005 | 0.572 +- 0.007 |
| DeepDDS-DR (WL k=3) | Proposed | 0.643 +- 0.008 | 0.563 +- 0.006 | 0.919 +- 0.006 | 0.705 +- 0.015 |
| DeepDDS-DR (SP) | Proposed | 0.634 +- 0.015 | 0.569 +- 0.007 | 0.917 +- 0.004 | 0.708 +- 0.025 |
| MatchMaker-DR (WL k=3) | Proposed | 0.666 +- 0.008 | 0.577 +- 0.005 | **0.941 +- 0.005** | 0.693 +- 0.014 |
| MatchMaker-DR (SP) | Proposed | **0.668 +- 0.014** | **0.581 +- 0.004** | 0.938 +- 0.004 | **0.730 +- 0.024** |

**Table 6:** Table of results with DR models utilising embeddings learned over the union of all drugs across the 4 datasets. We report the average AUROC on the hold out test set with standard deviations from 5 seeded random splits. Bolded numbers indicate best performing model for each dataset.

| Model | Year | DrugCombDB | DrugComb | DrugbankDDI | TwoSides |
|---|---|---|---|---|---|
| DeepDDS | 2021 | 0.791 +- 0.005 | 0.697 +- 0.002 | 0.988 +- 0.001 | **0.944 +- 0.001** |
| MatchMaker | 2021 | 0.788 +- 0.002 | **0.720 +- 0.003** | **0.991 +- 0.001** | 0.928 +- 0.001 |
| DROnly (WL k=3) | Proposed | 0.762 +- 0.002 | 0.652 +- 0.001 | 0.793 +- 0.002 | 0.909 +- 0.003 |
| DROnly (SP) | Proposed | 0.712 +- 0.002 | 0.615 +- 0.004 | 0.708 +- 0.004 | 0.796 +- 0.008 |
| DeepDDS-DR (WL k=3) | Proposed | **0.802 +- 0.002** | 0.700 +- 0.001 | 0.988 +- 0.001 | **0.944 +- 0.001** |
| DeepDDS-DR (SP) | Proposed | 0.785 +- 0.002 | 0.693 +- 0.002 | 0.983 +- 0.006 | **0.944 +- 0.001** |
| MatchMaker-DR (WL k=3) | Proposed | 0.783 +- 0.005 | 0.713 +- 0.004 | **0.991 +- 0.001** | 0.929 +- 0.001 |
| MatchMaker-DR (SP) | Proposed | 0.784 +- 0.005 | 0.714 +- 0.004 | **0.991 +- 0.001** | 0.929 +- 0.002 |

- If $d_i$ and $d_j$ are in $A$, this is a training observation.
- If $d_i$ and $d_j$ are from different sets, this is a test observation.
- If $d_i$ and $d_j$ are in $B$, this is a test observation.

4. This ensures that a drug pair scoring model never sees and instance of a drug from set $B$, which is also distinctly different from the training drugs in $A$ by way of Tanimoto similarity.

5. As an arbitrary choice we have chosen set $A$ to be the larger set of drugs after the clustering.

The effects of the above operations and the sizes of the different drug sets and the resulting train-test sets of triples is reported in Table 4. We trained and evaluate each of the models using the same experimental setup as in Section 4 of the main manuscript and report the results in Table 5. Firstly, we see that the task is indeed more challenging as the train-test splits ensures a given drug-pair scoring model never sees drugs from set $B$. This lowers the performance across methods as compared to random train-test splits in the main paper. The results also show that the distributed representations can help each of the methods perform better across the different drug pair scoring tasks in this more challenging setting. For both MatchMaker and DeepDDS the distributed representations improve performance or at least do not decrease the performance significantly. Increases in performance are particularly strong for DeepDDS in DrugCombDB and TwoSides. One observation to be made is that the DROnly performance may be a good indicator of potential gains to be made when the drug embeddings are incorporated into other models. The best performing method in general is MatchMaker-DR (WL or SP) which is a fortunate observation as it is considerably cheaper to train than DeepDDS. These results further suggest the positive impact the incorporation of distributed representations of graphs has on drug pair scoring models.

## C.2 Experiments with distributional shift in substructure patterns

As distributed representations are necessarily learned in a transductive manner we believe that the most realistic approach of using the distributed representations in transfer settings would be to learn the embeddings of all the drugs in the DrugComb, DrugCombDB, DrugbankDDI and TwoSides datasets. We then performed the same evaluation with the same experimental setup as in the main

part of the paper with the random train-test splits using these new embeddings and report the results in Table 6. The results indicate more variable positive results as compared to distributed representations learned on each subset of drugs separately. Specifically, we can see a stronger increase in performance for DeepDDS when using distributed representations in DrugCombDB than in Table 2, and generally performance increases for DeepDDS across datasets. Decreases in performance as seen on MatchMaker in DrugCombDB and DrugComb are the same as in Table 2. The distributed representations do not hurt MatchMaker on DrugbankDDI and TwoSides. These results indicate that the neural drug pair scoring models in general are able to extract useful features for their end-to-end task from the incorporation of distributed representations.

