# OpenReview forum: "Distributed Representations of Graphs for Drug Pair Scoring"
_logconference.io/LOG/2022/Conference — LoG 2022 Poster_

### Official Review · Reviewer_57SC · 2022-10-19

**Overall Score:** 6
**Confidence:** 3

**Review:**

(1) This paper explores how to incorporate distributed representations of graphs of drugs in the drug pair scoring tasks. Considering the real-world constraints and prior knowledge, the patterns would not be extremely huge space. Furthermore, the authors propose an approach to learn the distributed representations. They provide the experiment results and test the embeddings on other models as well.

(2)
Strong:
     (a) It is novel to leverage the graph kernel and distributed representation learning on graphs in the drug pair scoring task;
     (b) It is convincing to see the effectiveness of incorporating the proposed approach into existing drug pair scoring pipelines, which shows the generalization.
     (c) Authors release the drug embeddings for selected datasets openly. I believe this would contribute to this community a lot.
Weak:
     (a) I am wondering about the algorithm efficiency to calculate the pair score using graph representation learning.
     (b) Presentation quality, such as figure qualify and notation presentation.
     (c) I am just a little bit concerned about the novelty of their techniques.

(3) Accept. The research question is very interesting and the proposed approach solves the question appropriately. Their work also demonstrates the generalization tested on other models and the open-released embeddings could contribute to this community to some extent.

(4) Authors explore an open research question. The drug pair scoring research question and their proposed solution match well. The proposed framework addresses this issue clearly. Their generalization is sound and convincing.

(5) How do you self-evaluate the originality of the loss function and the proposed framework from the technical perspective rather than the setting angle?

(6) I am wondering if there are any comparable baselines that can learn the embeddings. In addition, I am interested to see the algorithm's efficiency and robustness.

(7) 9 page track

---

### Official Review · Reviewer_cL75 · 2022-10-21

**Overall Score:** 6
**Confidence:** 3

**Review:**

**Summary**
This paper introduced a method for learning distributed representations of drugs. In the literature, there are two types of commonly used representations of drugs. The first type is the determinisitc representations that are motivated by domain knowledge, such as the SMILES, which is based on the 1D linear descriptor of molecules, and topology fingerprints, which are based on the 2D descriptors. The second type is data-driven, task-orientated representations that are learned by (supervised) machine learning models (e.g., graph neural network) in an end-to-end fashion. The representation method proposed in this work lies in between those two types - it learns dense, low-dimensional representations that capture the topology features from drug molecule data in an unsupervised way (called distributional representations).

The idea is similar to the word2vec method in NLP but generalized to graph-structured data. Although learning distributed representations of individual molecule graphs has been studied before (e.g., ref 29), this paper considered an extension case where the learned representations of individual drugs also reflect the pairwise relationships between drugs (e.g., drug synergy or polypharmacy effects and drug-drug interactions), thereby facilitating drug pair scoring. The authors showed that the representations can be readily incorporated into existing machine learning models and improved the prediction performance in several downstream tasks of drug pair scoring.

**Overall recommendation**
Although I think the proposed method is novel and interesting (see “strengths” below), I am inclined to vote for weak reject due to that the evaluations were not very comprehensive (see “weakness” below)

**Strengths**
- The problem formulation is novel. As opposed to previous studies that learn representations that only capture topology features of individual molecule graphs, this paper aimed to learn representations that additionally take into account the pairwise relationships between molecules. Explicitly modeling the pairwise relationships in the representation learning can potentially improve many link prediction-like tasks for drugs.
- The work potentially has many applications because distributed representations are easy to integrate with machine learning models (as compared to large pre-trained deep neural networks that require fine-tuning) and this work seems to be the only type of drug representation that capture pairwise drug relationships in the literature.
- This paper was well written and easy to follow.

**Weakness**
- The empirical results were not very impressive. When combined with other machine learning models, the improvements due to the developed distributional representations seemed marginal -- in the four tasks in Table 2, the proposed approach only improved the prediction performance on a task by ~0.02 (absolute AUROC improvement) and <0.001 on the other three tasks. And the 0.001 improvements were sometimes within the -/+std range.
- The generalizability of the developed representation was not well demonstrated. For example, it seems the training and test sets were split randomly (Line 292), and it is not clear whether the representations have good generalizability to predict for unseen drugs.

**Suggestions for improving the paper**
I feel the current work can be largely improved by strengthening the evaluations. An important aspect of demonstrating the usefulness of the developed representations is to evaluate their robustness and generalizability in prediction tasks. This is especially critical for distributional representations as they will be used in an unsupervised way as plug-in features for ML models.  Below I list a few suggestions that would help make the arguments of this paper more convincing.

- Creating a more challenging train/test split. Following up on one point listed in the “weakness” section, the authors can consider benchmarking their methods on a train/test split with controlled similarity. For example, we can ensure that the Tanimoto similarity between any pair of a training drug and a test drug is less than a threshold. This evaluation will simulate the scenarios where the model is used to predict novel drugs that it has not seen in the training data, a more challenging setting than the random split in the paper.
- Validating the prediction performance across tasks. The paper introduced four datasets to evaluate the methods. An interesting experiment would be training the representations on one task and evaluating them on another task. Since distributional representations are often pre-computed and used on a new task, it is important to know how they perform on the new dataset and whether its performance would be influenced by the distribution shift of vocabularies (i.e., substructures of molecular graphs). And if so, how much?

**Minor points**
- Line 201, j \in V should be p_j \in V?

========================================

**Updates after the paper revision period**: The authors have conducted additional evaluation experiments to address my comments in the "Suggestions for improving the paper". The new results strengthened the manuscript and demonstrated the good performance of the proposed method. I appreciate the authors' effort in adding those results in the short period of paper revision. I am very happy to **raise my score** from weak reject (5) to weak accept (6). I think the proposed method would be a novel contribution to the literature as it learns distributional representations of drug *pairs*, which is a new formulation that differs from many existing studies that only learn representations of *individual* drugs.

---

### Official Review · Reviewer_eEd5 · 2022-10-26

**Overall Score:** 6
**Confidence:** 4

**Review:**

This paper presents a novel methodology for learning and incorporating distributed representations of graphs within a unified framework for drug pair scoring. The experimental results show that the method improves the downstream performance across different drug pair scoring tasks. There may be some positive impact on the community.

Strength:

[1] The paper argues to learn distributed representations of graphs as a source of additional features, which is reasonable within drug pair scoring pipelines. This is a novel perspective for such problems.

[2] The related work is well-structured and basically made a comprehensive summary of the relevant research.

[3] The experimental results show that current models with the incorporation of distributed representations improve performance across different tasks and datasets. It demonstrates that the method has good scalability.

Weakness:

What I am most concerned about is that the motivation of this article is relatively appropriate, but the innovation of the method level seems to be only related to the loss function. It also lacks the basic theoretical analysis of mathematics. I think the author needs to consider strengthening this point.

---

### Official Review · Reviewer_hFwR · 2022-10-27

**Overall Score:** 8
**Confidence:** 4

**Review:**

This work is a 9 page research article which presents a technique to learn distributed representations of molecular graphs and how use them to improve the overall performance of machine learning pipelines where the underlying task involves drug pair scoring.

The strong points of this work are: the novelty; performance of the proposed technique; detailed description of the experimental setup; reproducibility.

The weak points of the work are: that some parts are left implicit (i.e. the dimensions of the feature vectors; if the molecular graphs have a different number of nodes/edges; the feature sets names are clear from the context but not specified directly and a reference between the values 'pj' and the substructure patterns).

My recommendation is to accept the work: it provides a method for improving the performance of a machine learning model in a drug pair scoring task. The proposed method allows to store the embeddings of the graph molecules in a distributed fashion and use them in any moment for several downstream tasks. Moreover, in the experimental results the authors show that the method described in this work actually improves the performance of the current SOTA models for drug pair scoring by a significant amount. The experimental setup is described in every detail and the results are fully reproducible and the results obtained as well as the embedding used are released as open source within the supplementary material.

I would ask the authors if they can provide some details about the models used to produce the embeddings as well as the overhead introduced in terms of computational resources to produce and store the distributed representations.


To improve the work I suggest to look at some typos (bold of h_i^{l-1} in eq. 1, DrugSynergy model should be renamed as DeepSynergy); adjust some inconsistencies in the notation (the index 'd' of the graphs is provided both as apex and as subscript); define properly the sets for which you are interested in computing the cardinality (i.e. Eq. 6 and the following equation) to avoid confusion. I would also separate Fig. 1 into two figures to highlight the different stages involved as well as magnify the font of the text in the figure (at the moment, a zoom might be required to read clearly the text). Also the examples provided in Fig. 1 can be moved into the caption of the figure to have a more clear overview of the proposed technique. Finally I suggest to merge the curves in Fig.2 and Fig.3 into a single figures with two subfigures and use different line style or markers to distinguish the dataset names as well as rescale the y axis properly to appreciate how the curve changes across the different epochs.

---

### Meta-Review · Area_Chair_QskH · 2022-11-15

**Confidence:** 4
**Recommendation:** Accept

**Meta Review:**

This paper presents a new technique to incorporate distributed representations of graphs for modeling drug-drug interactions. The authors conduct extensive evaluation and demonstrates clear improvement over existing baselines. Indeed, all the reviewers unanimously vote for acceptance and therefore this paper should be accepted.

---

### Decision · Program_Chairs · 2022-11-23

Accept (Poster)